# Anti-Pollution Activity, Antioxidant and Anti-Inflammatory Effects of Fermented Extract from *Smilax china* Leaf in Macrophages and Keratinocytes

**Yoo-Kyung Kim**  and Dae-Jung Kang *

MNHBIO Co., Ltd., 172 Dolma-ro, Bundang-gu, Seongnam-si 13605, Gyeonggi-do, Republic of Korea
* Correspondence: djkang@mnhbio.com; Tel.: +82-31-212-0677

**Abstract:** Air pollution has considerable effects on the human skin, showing that every single pollutant has a different toxicological impact on it. The oxidative stress that exceeds the skin's antioxidant capacity can lead to oxidative damage and premature skin aging by repeated air pollutant contact. In this study, according to the generalized protocol available to objectively substantiate the 'anti-pollution' claim, we evaluated several biomarkers after pollutants exposure in Raw 264.7 macro-phages and HaCaT keratinocytes to investigate the possibility of anti-pollution cosmetic material of fermented extract from *Smilax china* leaves (FESCL). FESCL decreased pollutants-induced luciferase activity in a dose-dependent manner, and FESCL significantly inhibited XRE-luciferase activity at a concentration of 1%. The $IC_{50}$ value of FESCL showed the same DPPH scavenging activity at 0.0625% as ascorbic acid, and the maximum DPPH scavenging activity (92.44%) at 1%. The maximum permissible non-cytotoxic concentrations of FESCL for a Raw 264.7 cell was determined to be 2%, where $PGE_2$ production of FESCL was inhibited by 78.20%. These results show the anti-pollution activity of FESCL against the pollutant-stimulated human living skin explants. In conclusion, we confirmed the anti-pollution potential of FESCL as one of the functional materials in cosmetic formulation.

**Keywords:** anti-pollution; anti-inflammation; antioxidant; fermentation; *Smilax china* leaf; *Lactobacillus* spp.



## 1. Introduction

Exposure to air pollution has been noted as closely associated with increased morbidity and mortality worldwide. Airborne pollutants are known to penetrate the human body through multiple routes, including direct inhalation and ingestion, as well as dermal contact, and also to cause well-documented acute and long-term effects on human health [1]. Air pollution is composed of a heterogeneous mixture of compounds emitted directly from pollution sources, including gases, low molecular weight hydrocarbons, persistent organic pollutants, (e.g., dioxins), heavy metals, (e.g., lead and mercury), and particulate matter (PM) [1]. Air pollution has considerable effects on the human skin, and it is generally known that every single pollutant has a different toxicological impact on it. Especially being the largest organ of the human body, as well as the boundary between the environment and the organism, the skin unsurprisingly is one of the major targets of air pollutants. Ref. [1] Even though the skin is equipped with an elaborate antioxidant defense system, including enzymatic and non-enzymatic, hydrophilic and lipophilic elements, the oxidative stress that exceeds the skin's antioxidant capacity, can lead to oxidative damage, premature skin aging, and eventually skin cancer by repeated air pollutant contact [2].

Recently, many studies reported potential explanations for outdoor air pollutants' impact on skin damage, focusing their interest, especially on airborne particulate matter (PM) and the ozone [3–5]. PM consists of mixtures of particles of different sizes and compo-sitions and is a major concern in the air of densely populated urban areas [6]. According to

previous studies, PM has been known to induce skin oxidative stress, producing reactive oxygen species (ROS), and lead to the secretion of pro-inflammatory cytokines (TNF-$\alpha$, IL-1$\beta$, and IL-8) [7]. Additionally, it was reported that PM could be a relevant risk factor for the development of atopic dermatitis, a chronic inflammatory skin disease, and even exacerbate this condition [8–10]. There has been an increasing interest in anti-pollution products that protect them from various environmental threats.

The genus *Smilax* belongs to the *Liliaceae* family and contains 350 species, and it is known to be widely distributed in the tropical and temperate zones throughout the world, especially in tropical regions of East Asia and South and North America [11]. Many of them have been used as medicinal herbs in East Asian countries. For example, *Smilax china* is commonly used in traditional Chinese medicine for the treatment of diuretics, rheumatic arthritics, detoxication, lumbago, gout, tumors, and inflammatory diseases [12]. Recent pharmacological studies reported that *Smilax china* exhibited anti-inflammatory activity in skin cells [13–16] and significant cytotoxicity against several tumor cell lines [17,18]. In Korea, *Smilax china* L. is called 'Manggae,' and its leaves have been used in wrapping a unique Korean rice cake, called 'Manggae-dduck' of Uiryong-gun. This indigenous 'Manggae' also showed a notable antioxidant activity, similar to foreign *Smilax china* leaf (SCL) [19]. Especially, Li et al. reported that *Smilax china* had positive effects on skin wound healing and skin barrier function by accelerating keratinocyte migration [20].

In this study, according to the generalized protocol available to objectively substantiate the 'anti-pollution' claim, several in vitro tests were investigated. In order to investigate the anti-pollution potential of FESCL as one of the functional materials in the cosmetic formulation, we evaluated several biomarkers after pollutants exposure in Raw 264.7 macrophages and HaCaT keratinocytes. These two cell lines were used according to the generalized protocol available to objectively substantiate the 'anti-pollution' claim.

## 2. Materials and Methods

### 2.1. Materials

The dust PM1648a used in this study is urban particulate matter and was purchased from the National Institute of Standards and Technology (NIST; Gaithersburg, MD, USA). It has an average particle size of 5.85 μm. Cadmium chloride (202908), ascorbic acid (A4544), DPPH (D9132), and dexamethasone (D4902) were purchased from Sigma-Aldrich (St. Louis, MO, USA). MRS broth was purchased from Difco™. *Smilax china* leaves (SCL) were collected in 2019 in Uiryong-gun, Korea.

Human keratinocytes (Human adult low calcium high temperature: HaCaT) cells and Raw 264.7 cells used in this experiment were presold by the Korean Cell Line Bank, and DMEM medium containing 1% Penicillin streptomycin (Gibco, Grand Island, NY, USA) and 10% FBS was used. Incubated in a $CO_2$ incubator adjusted to 37 °C, 5% $CO_2$.

### 2.2. Preparation of Standardized SCL Fermented Extracts

SCL was dried at 30–40 °C for 48 h and dried samples were pulverized into a powder having a particle size of 40 mesh using a stainless steel blender (RT-08; MHK., Seoul, Republic of Korea). One kg of pulverized SCL was soaked in 5 L of distilled water for 2 h and then extracted at 100 °C for 2 h. The extract was separated from the insoluble part by filtration with Watman No.1 paper. The extracts were centrifuged at 8000 rpm for 20 min, and the supernatant was concentrated to 1 L under reduced pressure, then it was kept at 4 °C until use. The 1% (w/v) of freeze-dried Lactobacillus bulgaricus (KCTC13554BP) and Lactobacillus reuteri (KCTC14022BP), which were pre-cultured in MRS media, were inoculated in 1 L of SCL concentrated extract. This culture was fermented at 37 °C for 16 h and 10 °C for 6 h, sequentially, resulting in preparing the fermented extract of the SCL sample for this study.

The quantification of quercetin was analyzed by high-performance liquid chromatography (HPLC) (Waters 2690; Waters Co., Milford, MA, USA) using a Capcell pak C18 MG column (Shiseido, Ginza, Japan) (4.6 × 250 mm, 5μm) and a UV detector (Waters 2487;

Waters Co.). Samples were eluted isocratically with 10% (*v/v*) acetonitrile containing 0.1% (*v/v*) potassium dihydrogen phosphate monobasic at a flow rate of 1 mL/min and detected at 340 nm. To avoid variations in activity for different preparations, a sufficient extract was obtained in one batch for use throughout the study. The content of the marker quercetin in FESCL was quantitated using high-performance liquid chromatography and the total flavonoid content was analyzed as the quercetin equivalent (QE) mg/g of FESCL. Results indicated that FESCL possessed 25.3 QE mg of quercetin per 1.0 g (Supplementary Figure S1).

### 2.3. Cytotoxicity Assay

A WST-1 assay was carried for measuring the FESCL cytotoxicity value, about $5 \times 10^5$ viable cells were added to each well of a 96-well tissue culture plate with medium containing 10% FBS (fetal bovine serum), 1% Penicillin/Streptomycin (P/S), and incubated overnight at 37 °C to allow cells to attach to wells of the 96-well cell culture plates. The medium was replaced with fresh serum-free and 1% Penicillin/Streptomycin (P/S). Cell viability was measured using the EZ-Cytox cell viability assay kit (Itsbio. Seoul, Republic of Korea). The cells were incubated for 2 h at 37 °C in a serum-free medium diluted with 1 kit reagent. Next, harvested cells resuspended in the media were carefully moved to an empty 96-well plate, and absorbance was measured using an ELISA at 450 nm. Due to the measure of the cytotoxicity of FESCL in heavy metals exposure, after overnight incubation of the cell culture, the medium containing the serum-free and 1% P/S was replaced and treated with 4 μg/mL of cadmium chloride on each culture plate. The treatment of FESCL (0.0625, 0.125, 0.25, 0.5, and 1%) was suspended in cells and maintained for 24 h to confirm the cytotoxic concentration ranges. The percent cytotoxicity was calculated as follows:

$$Cell\ viability(\%) = \frac{(OD\ value\ of\ treatment - OD\ value\ of\ blank)}{(OD\ value\ of\ control - OD\ value\ of\ blank)} \times 100$$

(*OD*: optical density at 450 nm. Control: only cadmium chloride treatment).

### 2.4. The DPPH Radical Scavenging Activity

The *DPPH* radical scavenging activity of FESCL was determined according to the method of You et al. [21]. After diluting FESCL by concentrations of 0.125 mL for each fermented extract, 0.125 mL of PM1648a were mixed with 0.625 mL of 0.1 mM *DPPH* in ethanol at 4 °C for 30 min. Then, the absorbance of the sample were measured at 520 nm by a spectrophotometer (UV-1601; Shimadzu, Kyoto, Japan). Radical scavenging activity was expressed as a percentage according to the following formula:

$$DPPH\ radical\ scavenging\ activity(\%) = \left\{ 1 - \left( \frac{sample}{control} \right) \right\} \times 100$$

### 2.5. XRE-Luciferase Activity

To assay the activity of XRE and ARE-containing promoters, cells were transfected with XRE-luciferase (XRE-Luc) (Stratagene, La Jolla, CA, USA) or ARE-luciferase (ARE-Luc) reporters (Add gene, MA, USA), and Renilla-luciferase plasmid (1μg) (for normalization) (Promega, Madison, WI, USA) using the DharmaFECT® Duo transfection reagent (Thermo Fisher Scientific, Waltham, MA, USA), according to the manufacturers' protocols [22]. At 24 h post-transfection, FESCL by concentrations and PM1648a were added to the cells for a 48 h treatment. The cells were harvested and luciferase activity was measured using the Dual Luciferase Assay system (Promega) on a LB953 luminometer (Berthold, Germany).

### 2.6. Measurement of Prostaglandin E₂ (PGE₂) Production

The PGE$_2$ concentration in the culture medium was measured by the PGE$_2$ immunoassay (ELISA) kit (Enzo Life Sciences, Farmingdale, NY, USA) following the manufacturer's protocol. In brief, culture supernatants from the Raw 264.7 cells treated with various concentrations of FESCL with or without PM1648a (100 μg/mL) for 24 h were placed in

96-well plates with standard reagents. Wells were incubated with $PGE_2$ conjugate liquid and monoclonal $PGE_2$ antibody liquid for 24 h at 4 °C. After 24 h of incubation, wells were washed three times with a wash buffer and incubated with a substrate solution for 1 h at 37 °C. Then, the reactions were blocked by adding a stop solution reagent in each well. The optical density was determined using an automated microplate reader at 405 nm.

### 2.7. Statistical Analysis

All results are expressed as the mean ± standard deviation (SD) of at least three experiments. Statistical analyses were performed using the Statistical Package for the Social Sciences (SPSS 20.0) software. A paired *t*-test for the independent samples was used for the statistical analysis of the data. Values of * $p < 0.05$ displayed statistical significance.

## 3. Results

### 3.1. Cytotoxicity of FESCL in Heavy Metals Exposure

We investigated the cytotoxic effect of FESCL in a HaCaT cell, and it was observed to reduce viability in a concentration-dependent manner (Figure 1a). The viability of a HaCaT cell was not reduced following treatment with 0.125–1% of FESCL, but the viability was significantly reduced from 2% of FESCL. Based on these results, non-cytotoxic concentrations of FESCL (0.125, 0.25, and 1%) were used in further experiments. In order to evaluate the effect of FESCL on skin cells by heavy metals exposure, we measured the cytotoxicity of the HaCaT cell by treating FESCL in the condition of exposing 4 µg/mL of cadmium chloride. The viability of the HaCaT cell was significantly ($p < 0.05$) increased in a FESCL concentration-dependent manner. Notably, the HaCaT cell showed the highest value in 0.5% of FESCL treatments ($p < 0.05$), compared with the not-treated control (Figure 1b). Taken together, these results indicated that FESCL played potential roles in somewhat protecting skin cells from damage by toxic heavy metals exposure.

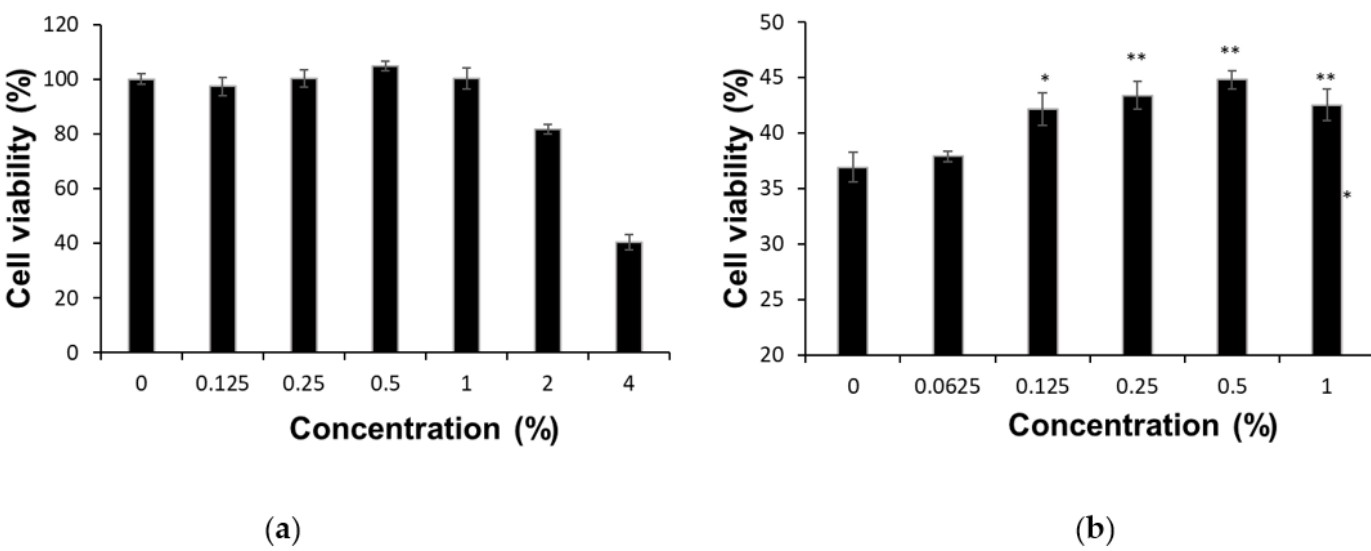

(a)  (b)

**Figure 1.** Inhibitory effect of FESCL cell cytotoxicity induced by treatment of Cadmium chloride (4 µg/mL) in HaCaT cells. (**a**) The cell viability of FESCL in HaCaT cells (**b**) The cell viability of FESCL in HaCaT cells with treatment of Cadmium chloride (4 µg/mL). The data are presented as the mean ± SD of three replicates. * $p < 0.05$ and ** $p < 0.01$.

### 3.2. Inhibitory Effect of FESCL on Air Pollutants

Xenobiotic-responsive elements (XREs) are the domains in the promoter region of some xenobiotic-responsive genes, and these gene expressions can be regulated through the interaction of the XRE and the dimer-containing aryl hydrocarbon receptor (AHR) [23]. The AHR is induced by polycyclic aromatic hydrocarbons (PAHs), and AHR dimerizes with ARNT and binds to xenobiotic-response elements which regulate the expression of

cytochrome P450. The oxidation of xenobiotics is important in dermatitis, and even more critical in the detoxification of carcinogens [24]. Accordingly, we evaluated the inhibition effect of FESCL on air pollutant (PM1648a) in the HaCaT cell by an XRE-luciferase activity (Figure 2). The cells were incubated with various concentrations of FESCL (0.25, 0.5, and 1%) and then exposed to the air pollutant (PM1648a-10 µg/mL). As evident in Figure 2, it was confirmed that FESCL decreased pollutant-induced luciferase activity in a dose-dependent manner, and FESCL significantly ($p < 0.001$) inhibited XRE-luciferase activity at a concentration of 1%. The results indicated that air pollutant activated the AHR-signaling pathway, but it appeared that FESCL decreased the activated signaling pathway, hence reducing intracellular XRE promotor activity due to PM1648a, which seems to be due to the cell damage defense activity of FESCL by the PM1648a stimulation.

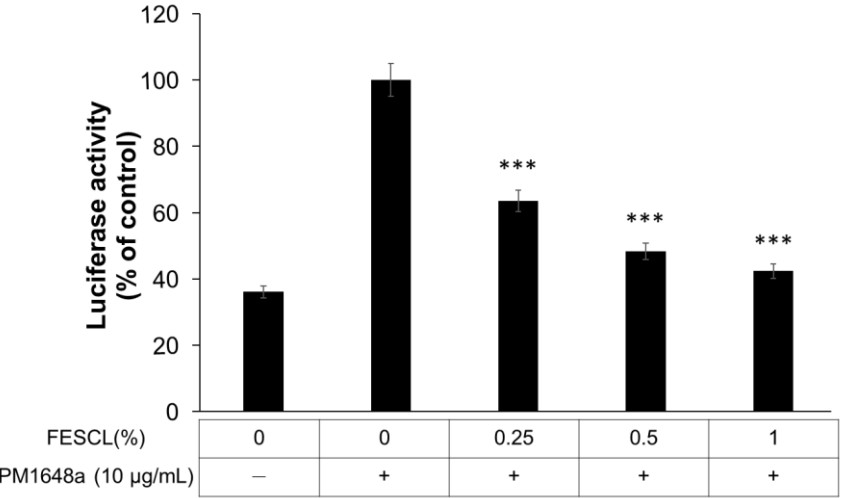

**Figure 2.** Luciferase activity of FESCL by the treatment of PM1648a (10 µg/mL) in HaCaT cells. The data are presented as the mean ± SD of three replicates. *** $p < 0.001$ vs. the PM1648a treatment control group.

### 3.3. Free Radical Scavenging Activity of FESCL on Air Pollutants

First, we tested the DPPH radical scavenging activity for PM1648a with ascorbic acid known to have high antioxidant activity as a positive control. As a result, it was confirmed that ascorbic acid exhibited concentration-dependent DPPH radical scavenging activity at a statistically significant level ($p < 0.05$). The entire treatment concentration range of ascorbic acid was from 0.313 to 10 µg/mL, and the $IC_{50}$ of ascorbic acid was 2.87 µg/mL. Then, the DPPH scavenging activity on air pollutant (PM1648a) was performed, respectively, to evaluate the antioxidant activity of FESCL (Figure 3). We also found that FESCL exhibited concentration-dependent DPPH radical scavenging activity at a statistically significant level ($p < 0.05$). The entire treatment concentration range was from 0.015 to 1% of FESCL, and the $IC_{50}$ of FESCL was 0.0618 µg/mL, and the highest DPPH scavenging activity was shown to be 92.44% at 1% concentration of FESCL. This result indicated that FESCL had a noticeable effect on the scavenging free radical and it had high antioxidant activity.

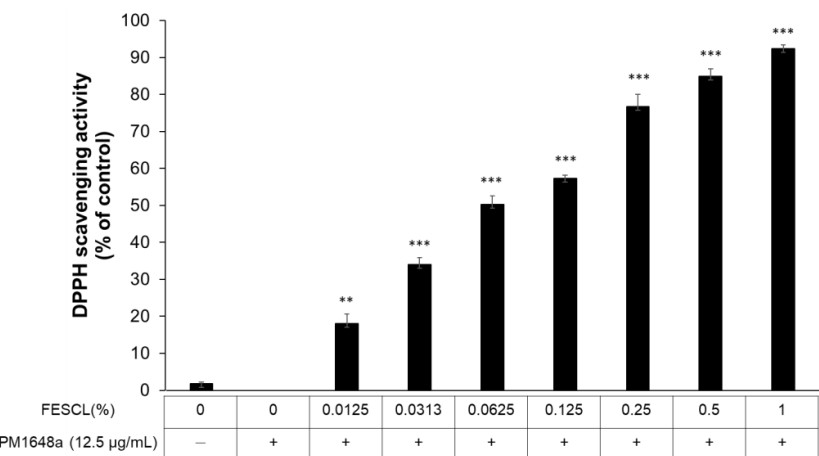

**Figure 3.** The DPPH radical scavenging activity of various concentrations of FESCL in HaCaT cells by treatment of PM1648a (12.5 μg/mL). All data were reported as the mean ± SD of three replicates. ** $p < 0.01$ and *** $p < 0.001$ vs. the PM1648a treated control group. DPPH; 2,2-diphenyl-1-picrylhydrazyl.

### 3.4. Anti-Inflammatory Activity of FESCL on Air Pollutants

The production of prostaglandin $E_2$($PGE_2$) is a representative inflammatory factor in immune cells, and the anti-inflammatory effect can be evaluated through the measurement of $PGE_2$ production inhibition activity [25]. Therefore, we measured the $PGE_2$ production inhibition activity of FESCL in Raw cell 264.7. After measuring the cytotoxicity level of fermented extract from SCL for Raw 264.7 cells, the FESCL inhibitory activity on $PGE_2$ production in the ones exposed to PM 1648a were evaluated. In addition, for the comparison of the positive control group, we tested the inhibition activity of $PGE_2$ production of 5μM dexamethasone-treated against the PM1648a stimulation and compared the anti-inflammatory effect of FESCL with dexamethasone (Figure 4). The non-cytotoxic concentrations of FESCL for the Raw 264.7 cell were by 2%, and the inhibition rate of $PGE_2$ production of FESCL was significantly ($p < 0.05$) increased in a concentration-dependent manner (0.125–2%). In particular, it was confirmed that about 78.20% of the $PGE_2$ production inhibited activity in the 2% concentration treatment group compared to the negative control group. $PGE_2$ levels in the medium were determined, as described in Materials and Methods. Cell viability by FESCL was also measured by a WST-1 based cell cytotoxicity assay, and there were non-cytotoxic concentrations of FESCL of 2%.

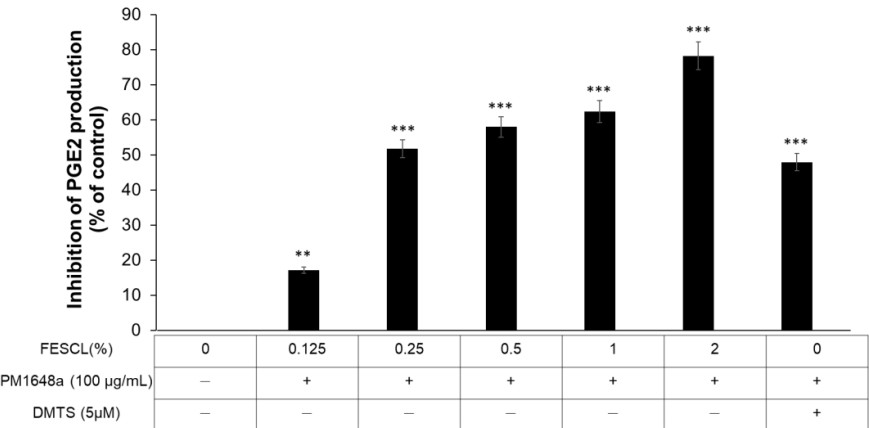

**Figure 4.** $PGE_2$ production inhibition activity of various concentrations of FESCL in Raw cell 264.7 by treatment of PM1648a (100 μg/mL). There were non-cytotoxic concentrations of FESCL by 2%. All data were reported as the mean ± SD of three replicates. ** $p < 0.01$ and *** $p < 0.001$ vs. the PM1648a treated control group. DMTS, dexamethasone.

## 4. Discussion

Cadmium (Cd) is one of the most concerning pollutants possessing high toxicity for both animals and plants. Cadmium is widely dumped into the environment through various anthropogenic activities such as mining, smelting, and the use of fertilizers [26]. Cadmium also affects cellular homeostasis and generates damage via complex mechanisms involving interactions with other metals and oxidative stress induction. Given it is one of the important pollutants which could damage human skin, we investigated a human keratinocyte cell line (HaCaT) as a model to study. In lines of experiments, we could confirm the significant protection activity of FESCL on the Cd exposed keratinocyte. This result was supported by Shi's previous report [27], in which *Smilax glabra* flavonoids extract could also reduce lead-induced cytotoxicity in HEK-293 cells stimulated with Pb.

As well, Urban PM is considered one of the most hazardous pollutants for human health [5], and in this study, the anti-pollution effects of FESCL were investigated by measuring antioxidant activity in the PM1648a-exposed HaCaT cells. As shown in Figures 2 and 3, the XRE-luciferase activity was significantly decreased by the concentration-dependent of FESCL, and the DPPH scavenging activity was significantly higher than the control group. Our results showed similar effect to the previous report related to the ethanol extract of *Smilax china*, but the FESCL appeared to be more effective compared to the ethanol extract of *Smilax china* [19]. According to Seo et al., the ethanol extract of *Smilax china* ($IC_{50}$ = 49.93 µg/mL) showed the highest DPPH radical scavenging activity in four solvents, whereas L-ascorbic acid (33 µg/mL) showed similar results. In contrast, in our study, the $IC_{50}$ value of FESCL was much lower than the $IC_{50}$ value of ascorbic acid ($IC_{50}$ = 0.0618 µg/mL). This means that the content of functional bioactive phytochemicals may be significantly enhanced by microbial fermentation.

Most of the previous studies have reported that *Smilax china* has bioactivities, such as anti-inflammatory, antioxidant, anticancer, and antimicrobial activities. However, these studies were mainly focused on the functionality of bioactive phytochemicals, not by fermentation extract, but the solvent extract of *Smilax china* [14–16]. Since the effects of FESCL on the antioxidant and anti-inflammatory effect had not been investigated yet, our study has great potential for improving skin health.

Since inflammatory reactions in the skin are a kind of defense reaction for protecting skin damage caused by physical irritation, chemicals, bacteria, and pollutants such as fine dust, the inflammatory reaction in the skin-activated macrophages produce inflammatory mediators such as $PGE_2$, and the inflammatory process could be further deepened [28,29]. We found the anti-inflammatory activity of FESCL by measuring the inhibition activity of $PGE_2$ production in Raw 264.7 cells. Feng et al. reported that flavonoid-enriched extract of *Smilax china* could affect suppressing inflammation in Raw 264.7 macrophages. Thus, the enhanced anti-inflammation of FESCL may be due to the increased phytochemical-like flavonoids or the synergetic effects of beneficial fermentation by *Lactobacillus*.

*Smilax china* a member of the *Smilacaceae* family, is widely distributed worldwide in tropical and temperate regions, especially in East Asia [15,19]. Previously, several studies have shown that *Smilax china* has been used in traditional medicine for the treatment of furunculosis, gout, tumors, and inflammation [12,14,27,30–32]. Recently, many studies have been discussing the antioxidant and anti-inflammatory effects of *Smilax china* leaves which has the presence of a significant amount of polyphenols [28,29,33]. The fermentation process itself yields beneficial effects through direct microbial action and the production of metabolites and other complex compounds [34]. During fermentation polyphenol compounds are metabolized and modified by fermenting organisms into other conjugates, glucosides, and/or related forms [34]. There were reports that fermentation positively confers organoleptic characteristics, and improves phenolic constituents and antioxidant activity [28,29]. Notably, FESCL showed high antioxidant and anti-inflammatory activity through a series of experiments using macrophages and keratinocytes after exposurepollutants. Taken together with findings and reports, the potential of FESCL as a functional material for cosmetics was expected to rise dramatically in the future.

## 5. Conclusions

In this study, we evaluated several biomarkers after pollutant exposure in Raw 264.7 macro-phages and HaCaT keratinocytes to investigate the possibility of anti-pollution cosmetic material of FESCL. FESCL decreased pollutant-induced luciferase activity in a dose-dependent manner, and FESCL significantly inhibited XRE-luciferase activity at a concentration of 1%. The $IC_{50}$ value of FESCL showed the same DPPH scavenging activity at 0.0625% as ascorbic acid, and the maximum DPPH scavenging activity (92.44%) at 1%. The maximum permissible non-cytotoxic concentrations of FESCL for a Raw 264.7 cell was determined to be 2%, where $PGE_2$ production of FESCL was inhibited by 78.20%. These results show the anti-pollution activity of FESCL against the pollutant-stimulated human living skin explants. In conclusion, we confirmed the anti-pollution potential of FESCL as one of the functional materials in cosmetic formulation.

**Supplementary Materials:** The following supporting information can be downloaded at: https://www.mdpi.com/article/10.3390/cosmetics9060120/s1, Figure S1: HPLC chromatogram of FESCL and calibration chromatogram of Quercetin as flavonoid marker.

**Author Contributions:** Conceptualization, Y.-K.K.; methodology, Y.-K.K.; validation, D.-J.K.; formal analysis, Y.-K.K.; writing-original draft preparation, Y.-K.K.; supervision, D.-J.K.; project administration, D.-J.K.; funding acquisition, D.-J.K. All authors have read and agreed to the published version of the manuscript.

**Funding:** This research received no external funding.

**Institutional Review Board Statement:** Not applicable.

**Informed Consent Statement:** Not applicable.

**Data Availability Statement:** Not applicable.

**Acknowledgments:** This work was supported by the Technological Innovation R&D Program (S2745843) funded by the Ministry of SMEs and Startups (MSS, Sejong City, Republic of Korea).

**Conflicts of Interest:** The authors declare no conflict of interest. Author Y.K. Kim has received research grants from MNHBIO Co., Ltd. Author D.J. Kang owns stocks in MNHBIO Co, Ltd.

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
