# Peer review of "Anti-Pollution Activity, Antioxidant and Anti-Inflammatory Effects of Fermented Extract from Smilax china Leaf in Macrophages and Keratinocytes"

_cosmetics, doi:10.3390/cosmetics9060120_

Round 1

Reviewer 1 Report

The present study has an interesting purpose; the design is well-conducted, and the results are promising.

Please, find below some comments and suggestions:

Because in macrophages was evaluated only the anti-inflammatory activity, I think that the title could be better as follows:  Anti-pollution activity, antioxidant and anti-inflammatory effects of fermented extract from Smilax china L. leaf in keratinocytes and macrophages.

Please, add a short phrase as a background for the abstract. 

Lines 9-10: To investigate the possibility of the anti-pollution cosmetic material of fermented extract from Smilax china L. leaves (FESCL), please justify why "cosmetic material"? Did you obtain a cosmetic formulation? Please, check and reformulate.

Line 10: Smilax china is better; please, check and correct the entire manuscript.

Lines 11 and 19: Raw/RAW - please, maintain the same style in the entire manuscript

Introduction:

It is well-presented in order, beginning with pathology, air pollutants, and the studied species. A short presentation of the plant parts used for therapeutic reasons may be welcomed.

Line 54: Liliaceae

Line 67:  ...were investigated - please, check

Lines 67-70: The authors showed the aim of their study as follows: "To investigate the possibility of anti-pollution cosmetic material of fermented extract from Smilax China L. leaves, we evaluated several biomarkers after pollutants exposure in Raw 264.7 macrophages and HaCaT keratinocytes". Why cosmetic material? Did you prepare a special cosmetical formulation with the studied fermented extract? Moreover, the anti-pollution potential is better. Please, check and correct. The authors could continue the aim of this study, presenting its novelty.

In section 2.1. "Materials," please, also add all reagents, cell lines, culture media, and kits with all providers.

Line 75:  Smilax china L., please, check and correct.

Lines 81-83: Please, reformulate the big phrase, avoiding repetition and grammar errors.

Line 84: What does 8,000 x g mean? And 54 Brix? (Line 85).

Lines 86-90: Please, reformulate the big phrase for a better understanding; maybe you could divide it.

Lines 91-97:  The HPLC determination is not well-described in "Materials and methods" - maybe it is previously performed? If the answer is positive, please, put it in the Supplementary Material and only remind in Discussion.

Line 116:  and treated with 4 μg/ml of cadmium chloride is better; please, check and correct.

Line 118: The percent of cytotoxicity was calculated as follows: - is better; please, check and correct.

Line 122: 1% antibiotic-antifungal mix is better. Please, mention the provider.

Line 158:  t-test

Lines 168-169: Due to measure the cytotoxicity of FESCL in heavy metals exposure, cadmium chloride 168 4μg/ml was treated on HaCaT cell. - please check the errors and reformulate.

Lines 169-172: Please reformulate this big phrase into two smaller ideas, avoiding repetition.

Lines 172-173: The results showed FESCL has inhibited for the cell  cytotoxicity in heavy metals exposure. - please reformulate for better understanding.

Figures 2-5: For better understanding, please specify the signification of all notations used, inclusively the controls in the figures' names. Figures 2b, 3, 4, and 5: it is better not to complicate the axes' names and to show only their signification, as in Figure 2a. All other data should be written in each associated text. Moreover, in the current form, all Figure captions are too long. Approximately 50% of them could be placed in the manuscript text and presented as results. 

Lines 233-235:  After measuring the cytotoxicity level of fermented extract from SCL for raw cell 264.7, the FESCL inhibitory activity on PGE2 production in the ones exposed to PM 1648a was evaluated - maybe it is better. Please, check and correct.

Lines 246-248: Please put in the text, before Figure 5, the following paragraph: "PGE2 levels in the medium were determined as described in Materials and Methods. Cell viability by FESCL was also measured by WST-1 based cell cytotoxicity assay, and there were non-cytotoxic concentrations of FESCL of 2%".

In "Discussion," please present in detail the cell lines selected for your study and justify your choice. Moreover, please compare your results with those of other previously published studies. If similar studies do not exist, you can mention this.

I suggest a particular section of "Conclusion" with a brief presentation of the essential data from this study and future perspectives. 

References:

Please, edit the References in MDPI Style.

As an overview, the manuscript needs extensive English language and style correction before publication. The writing style is cumbersome, with many extended phrases, repetitions, and grammatical errors, which can even change the statement's meaning. Such examples were previously provided. 

Author Response

We thank the editor and reviewers for their thoughtful comments regarding our manuscript. In the text that follows, we have responded to each of these comments in a point-by-point fashion (reviewer comments are in black font while our responses are in red font) and have made corresponding revisions to our manuscript. All revisions have been marked in red font in the revised manuscript. We believe these changes have significantly improved the manuscript and that it is now worthy of publication in Cosmetics

Reviewer 2 Report

It was a challenging and pleasant activity to review the article "Anti-pollution activity and antioxidant effects of fermented extract from Smilax China L. leaf in macrophages and keratino-cytes" by YooKyung Kim, and DaeJung Kang.

The manuscript presented, in my opinion, is good, and the results are an asset for progress in the area to which the manuscript refers.

However, I have a few comments to make:

_Introduction - I think that the authors should add in this chapter a paragraph about what is "Raw 264.7 macro-69 phages and HaCaT keratinocytes" and what their importance is for the study presented here.

_Line 91-97 - I think the authors should show this data in chapter 3, "Results". Please check. Still, these data are incomplete, and they must present the results of the validation of the method, such as the calibration line and the working range used.

_line 172 - I ask the authors to explain better the phrase "The results showed FESCL has inhibited for the cell cytotoxicity in heavy metals exposure.", bearing in mind that in Figure 2 (b), we only have 50% cell viability. To what extent does this percentage validate this statement? Please clarify.

_Line 252 - The title of the chapter "4. Discussion" should be changed to "4. Discussion and Conclusions", or it should be divided into two chapters, one for the discussion and one for the conclusion.

Author Response

(The authors gave the same response as above.)

Round 2

Reviewer 2 Report

Excellent review work. In my opinion, this revised manuscript is ready for publication.